

# A massive variable flavour number scheme for the Drell-Yan process

Rhorry Gauld

Nikhef, Science Park 105, NL-1098 XG Amsterdam, The Netherlands

r.gauld@nikhef.nl

## Abstract

The prediction of differential cross-sections in hadron-hadron scattering processes is typically performed in a scheme where the heavy-flavour quarks $(c, b, t)$ are treated either as massless or massive partons. In this work, a method to describe the production of colour-singlet processes which combines these two approaches is presented. The core idea is that the contribution from power corrections involving the heavy-quark mass can be numerically isolated from the rest of the massive computation. These power corrections can then be combined with a massless computation (where they are absent), enabling the construction of differential cross-section predictions in a massive variable flavour number scheme. As an example, the procedure is applied to the low-mass Drell-Yan process within the LHCb fiducial region, where predictions for the rapidity and transverse-momentum distributions of the lepton pair are provided. To validate the procedure, it is shown how the $n_f$-dependent coefficient of a massless computation can be recovered from the massless limit of the massive one. This feature is also used to differentially extract the massless $N^3LO$ coefficient of the Drell-Yan process in the gluon-fusion channel.

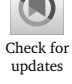

## 1   Introduction

The prediction of high-energy scattering processes which involve initial-state hadrons is crucial for understanding the physics of hadron collisions in both controlled environments (such as the LHC) as well as a range of naturally occurring scattering processes (such as those involving cosmic rays). In general, the starting point for the theoretical description that describes high-energy interactions in these collisions is a factorisation theorem [1] of the form

$$\frac{\mathrm{d}\sigma}{\mathrm{d}Q^2\,\mathrm{d}X} \sim \sum_{a,b} \int \mathrm{d}\xi_A \,\mathrm{d}\xi_B f_{a/A}(\xi_A, \mu) f_{b/B}(\xi_B, \mu) \times \mathrm{d}\hat{\sigma}_{ab}\left(\frac{x_A}{\xi_A}, \frac{x_B}{\xi_B}, Q; \alpha_s(\mu), \frac{\mu}{Q}\right).$$

This theorem separates the full scattering process into a partonic scattering process involving the scattering of the hadron constituents $a, b$ (e.g. quarks and gluons), and a set of parton distribution functions (PDFs) $f(x, Q)$ which describe the probability distribution of the internal content of the hadron as a function of hadron momentum-fraction and virtuality carried by the constituent particle. The energy-scale $Q$ denotes a representative scale of the scattering process, e.g. the dilepton invariant mass in the Drell-Yan (DY) process [2], and $X$ is a hadronic level observable such as the rapidity of the dilepton system.

A primary consideration when applying a factorisation theorem of this form is the treatment of heavy-flavour quarks (e.g. charm and beauty). For example, if/when it is a good approximation to consider these quarks as massless partons, or whether to retain the exact mass dependence of the heavy quarks. When treated as a massless parton, the heavy-flavour quark can be considered as an active parton in the perturbative evolution of PDFs as well as the strong-coupling $\alpha_s$. This approach is often convenient as, through this evolution, it allows to account for (to all orders) a class of logarithmic corrections to the scattering process of the form $\alpha_s^i \ln[m/Q]^j$ for $i \geq j$, where $m$ is the heavy-flavour quark mass. Instead, when considered as a massive parton, the impact of the heavy-quark mass can be incorporated exactly up to the known perturbative (fixed order) accuracy of the partonic cross-section $\mathrm{d}\hat{\sigma}$. This allows for the computation of the same logarithmic corrections as in the massless case outlined above (limited to fixed-order accuracy only), and in addition power corrections of the form $m/Q$ which are absent in the massless calculation.

Alternatively one can develop a scheme which combines these approaches, providing a uniform description of the scattering process across arbitrary energy scales—such a description is provided by a massive variable flavour number scheme. This topic has been studied in various contexts in the past [3–21], and with particular focus on the process of lepton-nucleon scattering [22–31]. In the latter case, it is well understood how to apply such a formalism to the description of nucleon structure functions. Due to the more rich structure of hadron-hadron scattering processes, the development and application of such a formalism is relatively less mature. It has been discussed for identified-hadron production [6, 8, 20, 32], processes with flavoured-jets [33–35], inclusive quantities [15, 36–39], exclusive quantities in the framework of SCET [40], and also within the context of Parton Showers [34, 41, 42].

The goal of this work is to revisit this topic for the production of colour-singlet processes, focussing on the neutral-current DY process. LHC measurements of this process have now

reached per-mille level accuracy [43], massless N$^3$LO predictions of such processes have now been obtained [44–46], and precise computations of the transverse-momentum spectrum are available at fixed-order [47,48] and beyond [49–55]. Given this progress, I believe it is important to unambiguously assess the importance of heavy-quark mass effects for fully differential collider physics predictions. To do so, I develop a method that allows to numerically extract (at the differential level) the contribution of massive power-corrections to the hadronic cross-section. The method is applicable to arbitrary processes, provided the considered observables are inclusive in QCD radiation and/or infrared and collinear safe. In anticipation of a measurement, and as an application and validation of the procedure, the low-mass Drell-Yan process is considered within the fiducial volume of the LHCb experiment. As a by-product of this work, I also show how the presented method can be used to obtain differential information on the DY cross-section, which is used to extract the N$^3$LO contribution to this process in the $gg$-channel.

## 2 (De)constructing the massive calculation

The general structure of the prediction of a differential hadronic-level cross-section involving a single massive quark can be written as

$$d\sigma^{\mathrm{M}} = d\sigma^{\mathrm{m=0,n_f}} + d\sigma^{\ln[\mathrm{m}]} + d\sigma^{\mathrm{pc}}. \tag{1}$$

The three contributions on RHS of Eq. (1) are: the $n_f$-dependent part of the calculation which is present when $m = 0$; those terms which depend logarithmically on $m$ which diverge in the limit $m \to 0$; and all remaining contributions that take the form of power corrections (labelled 'pc'), and which vanish in the limit $m \to 0$. The first two terms on the RHS of Eq. (1) are also present in a massless calculation (as they define the $m \to 0$ limit of the massive calculation), while the power corrections are uniquely described by the massive calculation. The core idea of this work is that the contribution $d\sigma^{\mathrm{pc}}$ can be numerically isolated by directly calculating all other terms appearing in Eq. (1). At fixed-order accuracy, this isolation procedure should be applicable to arbitrarily differential observables, provided they are inclusive with respect to QCD radiation and/or are infrared and collinear safe such that the zero-mass limit is well defined. This includes the differential description of a colour-singlet system (which will be the focus of this work), but also applies to processes involving hadronic jets (including those with identified flavour [56]). The application to identified hadron production is slightly different (due to the presence of final-state mass singularities), and has been discussed in the past [6, 8, 20, 32].

To illustrate how the procedure is performed, the neutral-current DY process (i.e. pp $\to \ell\bar{\ell} + X$) will be considered, and a description of how to evaluate each of the terms appearing in Eq. (1) is given. The current availability and the perturbative accuracy of these terms is also described.

**Massive computation, $d\sigma^{\mathrm{M}}$.** The cross-section $d\sigma^{\mathrm{M}}$ appearing on the LHS of Eq. (1) denotes the contribution from a single massive quark with mass $m$ to the hadronic scattering process. For the DY process, the presence of a massive quark alters the calculation starting at $\mathcal{O}(\alpha_s^2)$. The mass enters the calculation explicitly in subprocesses of the form $ab \to \ell\bar{\ell} + Q\bar{Q}$ (where $a, b$ denote massless partons and $Q$ the massive quark), but also implicitly enters the lower multiplicity subprocesses $ab \to \ell\bar{\ell}(+c)$ either in double-virtual corrections or through the definition of UV renormalisation counter-terms—see the Appendix of [57] for a detailed discussion. It should be clear that it is necessary to consider all contributions of the massive quark (whether they appear explicitly or not).

A massive calculation of the DY process is not available at $\mathcal{O}(\alpha_s^3)$. This requires perturbative ingredients, such as various two and three-loop corrections involving a closed massive fermion

loop, which are currently unknown.

**Zero mass computation,** $d\sigma^{m=0,n_f}$. When considered massless, the quark $Q$ still contributes to the same subprocesses as in the massive computation described above (but with zero mass). This contribution can be computed directly after extracting the $n_f$-dependent part of the massless partonic cross-section at this order. Due to the presence of single- and double-unresolved emissions in the differential calculation, this extraction must also be applied to the (integrated) subtraction/slicing terms which are required to regulate these emissions.

While first differential results for the massless DY cross-section at $\mathcal{O}(\alpha_s^3)$ have been presented [46] (relying on the NNLO QCD calculation for $Zj$ [47] reported in [52]), a careful (and lengthy) computation is required to extract the $\mathcal{O}(\alpha_s^3 n_f)$ component. A differential calculation of the massless $\mathcal{O}(\alpha_s^3 n_f)$ contribution to DY is therefore currently unavailable.

**Logarithmic computation,** $d\sigma^{\ln[m]}$. Provided that QCD inclusive and/or infrared- and collinear-safe observables are considered, the logarithmic dependence of the massive cross-section on the heavy-quark mass $m$ is of collinear origin. This behaviour is universal, and it can be described with knowledge of a set of decoupling relations which describe how parameters (e.g. $\alpha_s$ and PDFs) in a theory with a massive quark are mapped (at fixed-order accuracy) into an effective theory where that quark is treated as massless. With this information, it becomes possible to construct the logarithmic behaviour of the differential cross-section using only massless inputs.

This construction requires the decoupling relation for $\alpha_s$ (and $m$) which is known analytically to high perturbative-order [58], and also available with public software (see for example [59]). The corresponding relations for the PDFs are provided in the form of massive Operator Matrix Elements (OMEs, and denoted $\hat{A}_{ab}$) which describe the transition between the partonic states $a \to b$. The perturbative structure of these objects has been studied at great length in the past, and calculations of the massive OMEs are available at two-loop [10, 24, 60–65], and three-loop [66–70] order. Notably, these OMEs also define the matching conditions/decoupling relations which allow to construct a VFNS for PDFs—see for example Eq. (12-15) of [70] (and originally [24]).

To construct the logarithmic cross-section for the DY process, one has to consider convolutions of the form

$$\hat{A}_{ab}^{(i)} \otimes \hat{A}_{cd}^{(j)} \otimes d\hat{\sigma}_{bd \to \ell\bar{\ell}+X}^{(k),m=0} , \tag{2}$$

where the superscripts $(i-k)$ denote the perturbative order of the OMEs and the massless partonic scattering cross-section $(d\hat{\sigma}_{bd \to \ell\bar{\ell}+X}^{(k),m=0})$. All of the $\hat{A}_{ab}^{(i)}$ inputs required to construct $d\sigma^{\ln[m]}$ up to $\mathcal{O}(\alpha_s^2)$ (i.e. $i + j \leq 2$) have been presented in [24]. The results presented in [66–70] should also allow to extend this calculation to $\mathcal{O}(\alpha_s^3)$. For $(k) \geq 1$, the decoupling relation for the strong coupling $\Delta_{n_f}^{(i)}(\alpha_s)$ is also required, which can be applied as a multiplicative factor to Eq. (2). The perturbative expansion for $\Delta_{n_f}^{(i)}(\alpha_s)$ is reported in Eq. (20) of [59]. Working in the $\overline{\text{MS}}$ scheme for the strong coupling, and defining the heavy quark mass in the on-shell scheme (which is consistent with that of the OME calculation in [24]), the expansion is

$$\Delta_{n_f}(\alpha_s) = 1 + \frac{\alpha_s}{2\pi}\left(-\frac{1}{3}\ln\left[\frac{\mu^2}{m^2}\right]\right) + \left(\frac{\alpha_s}{2\pi}\right)^2\left(-\frac{7}{6} - \frac{19}{6}\ln\left[\frac{\mu^2}{m^2}\right] + \frac{1}{9}\ln^2\left[\frac{\mu^2}{m^2}\right]\right) + \mathcal{O}(\alpha_s^3). \tag{3}$$

Here, $\alpha_s$ denotes the strong coupling defined in the massless scheme (where the heavy quark is included in the running) at the scale $\mu$. Up to $\mathcal{O}(\alpha_s^2)$, the relevant expansion which is required to build the (partonic) logarithmic cross-section is

$$d\hat{\sigma}_{q\bar{q}}^{\ln[m],(2)} = \left(\hat{A}_{qq,Q}^{(2)} + \hat{A}_{\bar{q}\bar{q},Q}^{(2)}\right) \otimes d\hat{\sigma}_{q\bar{q}}^{(0)} + \Delta_{n_f}^{(1)}(\alpha_s) \cdot d\hat{\sigma}_{q\bar{q}}^{(1)} ,$$

$$d\hat{\sigma}_{gg}^{\ln[m],(2)} = \hat{A}_{gQ}^{(1)} \otimes \hat{A}_{g\bar{Q}}^{(1)} \otimes d\hat{\sigma}_{Q\bar{Q}}^{(0)} + \hat{A}_{gQ}^{(1)} \otimes \left(d\hat{\sigma}_{Qg}^{(1)} + d\hat{\sigma}_{gQ}^{(1)}\right) + \left(Q \leftrightarrow \bar{Q}\right) . \tag{4}$$

Constructed in this way (i.e. using massless inputs) the logarithmic calculation will also contain those terms which are independent of $m$. They are generated by the constant terms contained in $\hat{A}_{ab}$ and $\Delta_{n_f}(\alpha_s)$—i.e. those which define the de-coupling across heavy-flavour thresholds in a variable flavour number scheme. It is necessary to account for these terms as they are part of the massive calculation (i.e. they appear on the LHS of Eq. (2)), but are not generated when $d\sigma^{m=0,n_f}$ is computed with inputs (PDFs and $\alpha_s$) defined in the massive scheme (e.g. $n_f^{max} = 4$ for the $b$-quark).

## 3 Heavy-quark mass slicing to $\mathcal{O}(\alpha_s^3)$

Following from the discussion in the previous Section, it is clear that all ingredients required to extract the power corrections for the DY process are known up to $\mathcal{O}(\alpha_s^2)$. This extraction is achieved numerically by evaluating the first three terms appearing in Eq. (1) and solving for $d\sigma^{pc}$. To validate this procedure, is it important to test that the extracted power corrections vanish in the limit $m \to 0$, which can be done by performing the extraction for decreasing values of $m$. Extracted in this way, the power corrections will only vanish provided that $d\sigma^{\ln[m]}$ reproduces the logarithmic behaviour of the massive cross-section in the limit $m \to 0$, and that the calculation of the constant term $d\sigma^{m=0,n_f}$ is correct.

Viewed in another way, one can also use the small-mass limit to numerically extract $d\sigma^{m=0,n_f}$ when it is unknown. This can be achieved if both massive and logarithmic calculations are known at the desired perturbative order, by performing a fit to the constant difference $\left(d\sigma^M - d\sigma^{\ln[m]}\right)$ in the limit $m \to 0$. Once this constant is known, the power corrections can also be extracted at the physical value of the heavy-quark mass. In practice, this corresponds to a global slicing method, where the heavy-quark mass parameter $m$ acts as a collinear regulator.

This technique is noteworthy, as it can be used to extract differential results for the DY cross-section at $\mathcal{O}(\alpha_s^3)$ in the gluon-fusion channel. This is possible because both the massive and logarithmic calculations are available at this order. The massive calculation is simply the NLO QCD correction to the subprocess $gg \to \ell\bar{\ell}Q\bar{Q}$ which can be obtained with automated codes such as aMC@NLO [71,72], and the logarithmic cross-section can be constructed using the two-loop OMEs given in [24]. The (partonic) logarithmic cross-section at $\mathcal{O}(\alpha_s^3)$ is constructed following the same procedure which resulted in Eq. (4), but expanded to one order higher. At this order new technical features are encountered, which are briefly discussed below.

At $\mathcal{O}(\alpha_s^3)$, it is necessary to consider convolutions of the form

$$\hat{A}_{gQ}^{(1)} \otimes d\hat{\sigma}_{Qg \to \ell\bar{\ell}+X}^{(2),m=0}, \tag{5}$$

where $d\hat{\sigma}_{Qg \to \ell\bar{\ell}+X}^{(2),m=0}$ is the second-order massless partonic cross-section and $\hat{A}_{gQ}^{(1)}$ the one-loop OME in the $Qg$ channel. The partonic cross-section appearing in this convolution contains contributions from double-real, real-virtual, and double-virtual phase-space configurations—with (integrated) subtraction terms appearing at each level. It is therefore necessary to convolute $\hat{A}_{gQ}^{(1)}$ with all terms at all levels, which at the real-virtual and virtual-virtual level includes iterated convolutions with integrated subtraction terms. To better clarify this issue, it is sufficient to consider the construction of the logarithmic cross-section at finite $p_{T,\ell\bar{\ell}}$ values where the partonic cross-section $\hat{\sigma}_{Qg \to \ell\bar{\ell}+X}^{(2),m=0}$ effectively becomes NLO accurate (there can be maximally a single unresolved QCD emission at this order). Practically, in this case, one has to consider

terms of the form

$$+\hat{A}^{(1)}_{gQ} \otimes \left( \mathrm{d}\hat{\sigma}^{R,m=0}_{Qg\to\ell\bar{\ell}+X} - \mathrm{d}\hat{\sigma}^{Rs,m=0}_{Qg\to\ell\bar{\ell}+X} \right)$$
$$+\hat{A}^{(1)}_{gQ} \otimes \left( \mathrm{d}\hat{\sigma}^{V,m=0}_{Qg\to\ell\bar{\ell}+X} - \mathrm{d}\hat{\sigma}^{Vs,m=0}_{Qg\to\ell\bar{\ell}+X} \right), \tag{6}$$

where the superscripts $R, Rs, V, Vs$ denote real, real-subtraction, virtual, and virtual-subtraction terms which are present in a differential NLO calculation. The convolution with $\hat{A}^{(1)}_{gQ}$ must be applied to all terms. As the virtual-subtraction cross-section contains mass-factorisation terms and integrated subtraction terms which are distributions in a collinear variable, the convolution of these terms with $\hat{A}^{(1)}_{gQ}$ leads to the aforementioned iterated convolutions. To obtain a result which is valid at all $p_{T,\ell\bar{\ell}}$ values, the above procedure is applied to the NNLO accurate partonic cross-section.

At $\mathcal{O}(\alpha_s^3)$, it is also important to consider the scheme dependence of the various inputs which enter the construction in Eq. (2). For example, the OME calculation of [24] is valid when the input PDFs are renormalised in the $\overline{\mathrm{MS}}$ scheme with $n_f - 1$ light flavours while the strong coupling is renormalised with $n_f$ light flavours. Also the first-order partonic cross-section $\mathrm{d}\hat{\sigma}^{(1)}_{gQ}$ appearing in line 2 of Eq. (4) is defined with $n_f - 1$ light flavours (i.e. for $\alpha_s$ and the gluon PDF which is not convoluted with the OME). Practically, a scheme conversion is applied such that all inputs are defined with inputs (PDFs and $\alpha_s$) that are defined in the $\overline{\mathrm{MS}}$ scheme with $n_f$ light flavours. The relevant scheme conversions are

$$\alpha_s^{[n_f-1]} = \alpha_s^{[n_f]} \left( 1 + \Delta^{(1)}(\alpha_s^{[n_f]}) \right) + \mathcal{O}(\alpha_s^2),$$
$$g^{[n_f-1]}(x,\mu_F^2) = g^{[n_f]}(x,\mu_F^2) - \frac{\alpha_s^{[n_f]}}{2\pi} \hat{A}^{S,(1)}_{gg,Q} \otimes g^{[n_f]}(x,\mu_F^2) + \mathcal{O}(\alpha_s^2), \tag{7}$$

with the definition

$$\hat{A}^{S,(1)}_{gg,Q}\left(z,\mu_F^2/m^2\right) = -\delta(1-z)\frac{1}{3}\ln\left[\frac{\mu_F^2}{m^2}\right]. \tag{8}$$

Notice that when this conversion is applied to equal powers of $\alpha_s$ and the gluon PDF, the dependence on $m$ vanishes and a logarithm of the ratio $\mu_F/\mu_R$ remains.

## 4 Intrinsic charm contributions

So far, the discussion has implicitly assumed that the contribution from massive initial-state quarks is absent. This is consistent with the assumption that there are no intrinsic heavy-flavour PDFs, which is the set-up of most modern global PDF fitting groups. In contrast, the NNPDF collaboration have relaxed this assumption [73] (see also [74]), and now fits for an intrinsic charm quark PDF as part of the nominal fit. In this case, the formalism outlined above can also be applied to extract the massive power-corrections associated to initial-state charm quarks. This is done in this work, extending the previous results for DIS [30,75] and inclusive observables [21], to the fully differential level. This requires the use of the OMEs for massive initial states originally computed in [4] which have also been presented in the Appendix of [21].

As an aside, I note that the general factorisation theorem for the computation of hadronic-level cross-section predictions involving massive-initial state quarks is known to be violated at $\mathcal{O}(\alpha_s^2)$. This topic has been studied in the past [76–87], and has received recent attention in [88] in the context of the Drell-Yan process. A deeper theoretical understanding of factorisation theorems for massive-initial states remains desirable today.

# 5  Constructing the M-VFNS

The massive variable flavour number scheme (M-VFNS) is constructed by combining a massless calculation with that of the massive power-corrections outlined above, for differential predictions, according to

$$d\sigma^{\text{M-VFNS}} = d\sigma^{\text{m}=0} + \sum_{i=c,b,\dots}^{n_f^{\max}} d\sigma_i^{\text{pc}}. \tag{9}$$

In this matching formula, the first term $d\sigma^{\text{m}=0}$ is the massless computation (i.e. that in a zero mass variable flavour number scheme with $n_f^{\max}$ flavours, ZM-VFNS), and the second term denotes the power corrections which are obtained by re-arranging Eq. (1). The power corrections can be evaluated separately for each of the heavy-flavour quarks (at higher-orders, one could also extend the formalism to deal with the presence of two-mass contributions). The master formula Eq. (9) is similar to those which have been presented for DIS Structure Functions (e.g. [29]), where $d\sigma^{\text{pc}}$ is often written as the difference $d\sigma^{\text{pc}} = \left(d\sigma^M - d\sigma^{M\to 0}\right)$.

    With respect to either a massive or a massless approach, the benefits of this construction are that a resummation of a class of collinear logarithms involving the heavy-quark mass $m$ (through PDF and $\alpha_s$ Renormalisation Group evolution) are included to all orders, and the exact heavy-quark mass dependence is included to the fixed-order accuracy to which $d\sigma_i^{\text{pc}}$ is known.

    Details of the computational set-up used for this work are provided in the following Section, before providing a numerical validation of the procedure and phenomenological results relevant for a measurement by the LHCb collaboration.

# 6  Computational set-up

**Theoretical implementation.** The predictions of the various differential cross-sections which enter the construction of the M-VFNS for the DY process are provided with a specialised Monte Carlo programme. It was originally purposed to enable the construction of a M-VFNS for the pp $\to$ Z + $b$-jet process [35]. The programme has since been extended to contain all ingredients which are required for the computation of the process pp $\to \ell\bar{\ell} + X$ up to $\mathcal{O}(\alpha_s^2)$, which may involve involve massless or massive heavy-flavour QCD partons. This includes those axial contributions arising due the presence of heavy-quark triangle diagrams, see for example [89–91]. Processes involving massive initial states are instead limited to $\mathcal{O}(\alpha_s)$.

    These computations are performed using a combination of Dipole subtraction [92] to treat the presence of single unresolved emissions (see [18, 93] for massive initial-states), and N-jettiness slicing for double unresolved emissions [94]. This implementation relies on many existing results, which include: amplitudes [57, 90, 91, 95–98]; N-jettiness inputs [99–106]; several OpenLoops libraries for tree-level amplitudes [107]; as well as a number of results manually computed with the aid of `FeynArts` [108] and `FormCalc` [109]. Beyond fixed-order, the programme also facilitates the computation of resummed predictions of the $p_{\text{T},\ell\bar{\ell}}$ spectrum at NNLL accuracy using a combination of results from [110–112]. The numerical integration of all contributions in the Monte Carlo programme are performed with the `VEGAS` algorithm as implemented in `CUBA` [113].

**Numerical inputs.** All predictions are provided with the NNPDF3.1 NNLO PDF set [73] with $\alpha_s(M_Z) = 0.118$ (with $n_f^{\max} = 5$), where the PDF and $\alpha_s$ values are accessed via LHAPDF [114]. These PDFs are used as an input to all calculations, which requires the application of a renormalisation scheme change to some of the inputs which enter the $\mathcal{O}(\alpha_s^3)$ calculation in the

$gg$-channel. The use of these PDFs grids (and the corresponding $\alpha_s$ values) practically defines how the resummation is implemented within the first term appearing in Eq. (9). The values of the on-shell heavy quark masses in this PDF set are $m_{c,b,t}^{\text{pdf}} = 1.51, 4.92, 172.5$ GeV. It should be noted that the boundary condition for the PDF set is defined at $Q_0 = 1.65$ GeV, which is larger than $m_c^{\text{pdf}}$. This information is relevant as it is used to derive the the static charm-quark PDF $f_c(x)$, according to the de-coupling relations calculated in [4].

All calculations are performed in the Complex Mass Scheme [115], with Electroweak inputs defined in the $G_\mu$-scheme following [107]. The following values for the numerical inputs are used $M_Z^{\text{os}} = 91.1876$ GeV, $\Gamma_Z^{\text{os}} = 2.4952$ GeV, $M_W^{\text{os}} = 80.379$ GeV, $\Gamma_W^{\text{os}} = 2.085$ GeV, and $G_\mu = 1.16638 \times 10^{-5}$ GeV$^{-2}$. The massless DY computations at $\mathcal{O}(\alpha_s^2)$ use the N-jettiness slicing method with a technical parameter of $\tau_{\text{cut}} = 10^{-3}$ GeV. Such a small value was chosen (at substantial CPU cost) to suppress the impact of missing power corrections beyond those included via [106] (see also [116] for a recent discussion).

For the results shown in the following Sections, an uncertainty due to the impact of missing higher-order corrections is assessed by varying the values of $\mu_R$ and $\mu_F$ by a factor of two around the dynamical scale $\mu_0 \equiv E_{\text{T},\ell\bar{\ell}}$ (the transverse mass of the dilepton pair), with the constraint that $\frac{1}{2} \leq \mu_F/\mu_R \leq 2$. When the M-VFNS is constructed according to Eq. (9), the scale uncertainties are correlated between the power corrections and the massless computations. Where shown, PDF uncertainties have been obtained from individual replica predictions ($i$) calculated in the following way:

$$\mathrm{d}\sigma_i = K \, \mathrm{d}\sigma_i[\mathcal{O}(\alpha_s)], \qquad K = \frac{\mathrm{d}\sigma_0[\mathcal{O}(\alpha_s^2)]}{\mathrm{d}\sigma_0[\mathcal{O}(\alpha_s)]}. \tag{10}$$

That is to say that a differential K-factor is calculated for the central PDF member at $\mathcal{O}(\alpha_s^2)$, and then applied to each of the individual replica cross-sections which are computed at $\mathcal{O}(\alpha_s)$. **LHCb fiducial definition.** In anticipation of a measurement of the process $pp \to \ell\bar{\ell} + X$ by the LHCb collaboration at $\sqrt{S} = 13$ TeV, the procedure outlined in the previous Section is both validated and applied in the fiducial region of the LHCb experiment. The predictions will be performed double differentially with respect to the invariant mass of the dilepton pair within the range $m_{\ell\bar{\ell}} \in [4, 200]$ GeV and either the transverse momentum ($p_{\text{T},\ell\bar{\ell}}$) or rapidity ($y_{\ell\bar{\ell}}$) of the dilepton pair. The following set of cuts are applied to the charged leptons: $p_{\text{T},\ell} > 1.5$ GeV, $|p_\ell| > 20$ GeV, $2.0 < \eta_\ell < 4.5$.

Predictions have been generated at fixed-order accuracy for both $p_{\text{T},\ell\bar{\ell}}$ and $y_{\ell\bar{\ell}}$ distributions in 20 invariant mass bins (a total of 400 bins). In this work, I have chosen to present the results within the invariant mass region of $m_{\ell\bar{\ell}} \in [12.5, 13.5]$ GeV. This region is of phenomenological interest as it provides sensitivity to the input PDFs at small-$x$, without being overwhelmed by perturbative uncertainties which grow in the very low $m_{\ell\bar{\ell}}$ regime. At the same time, this invariant mass region is sufficiently small that the fixed-order predictions within the range of $p_{\text{T},\ell\bar{\ell}} \in [2.5, 11.0]$ GeV (which will be the focus of the $p_{\text{T},\ell\bar{\ell}}$ measurement) are expected to be reliable. At higher values of $m_{\ell\bar{\ell}}$, a resummation of Sudakov logarithms of the form $\ln[p_{\text{T},\ell\bar{\ell}}/m_{\ell\bar{\ell}}]$ is necessary.

The numerical validation of the theoretical procedure (to follow) will be performed in the following kinematic regimes:

$$
\begin{aligned}
\text{Fiducial}: \quad & p_{\text{T},\ell\bar{\ell}} \text{ inclusive}, \\
p_{\text{T},\ell\bar{\ell}}^{\text{low}}: \quad & p_{\text{T},\ell\bar{\ell}} \leq 2.5 \text{ GeV}, \\
p_{\text{T},\ell\bar{\ell}}^{\text{high}}: \quad & p_{\text{T},\ell\bar{\ell}} \geq 2.5 \text{ GeV}.
\end{aligned}
\tag{11}
$$

The inclusion of this additional restriction in $p_{\text{T},\ell\bar{\ell}}$ is relevant as this defines the bin edge of the on-going $p_{\text{T},\ell\bar{\ell}}$ measurement. Clearly, the Fiducial volume is the sum of the latter contributions.

# 7 Numerical validation and $\mathcal{O}(\alpha_s^3)$ results

**Validation at $\mathcal{O}(\alpha_s^2)$.** To first validate the procedure, the small-mass limit of the massive cross-section $\mathrm{d}\sigma^{\mathrm{M}}$ appearing in Eq. (1) is considered within the LHCb Fiducial region. This contribution is computed at $\mathcal{O}(\alpha_s^2)$ and is compared to the logarithmic cross-section $\mathrm{d}\sigma^{\ln[m]}$ at the same order. In both cases, the scales are set to $\mu_{\mathrm{F}} = \mu_{\mathrm{R}} = E_{\mathrm{T},\ell\bar{\ell}}$ and the contribution from $b$-quarks are considered. The results for $c$-quarks are qualitatively similar, and differ in magnitude due to the coupling of the quarks with the exchanged gauge-boson.

The results are shown in Fig. 1 with a breakdown into $q\bar{q}$- and $gg$-initiated channels ($q$ indicating a light-flavour quark). In the lower-panel, the cross-section difference $\left(\mathrm{d}\sigma^{\mathrm{M}} - \mathrm{d}\sigma^{\ln[m]}\right)$ is shown alongside the direct calculation of $\mathrm{d}\sigma^{\mathrm{m=0,n_f}}$. The direct and indirect (obtained via the $m \to 0$ limit) methods of calculating $\mathrm{d}\sigma^{\mathrm{m=0,n_f}}$ coincide, confirming the structure of the massive calculation presented in Eq. (1).

It is important to highlight that the massive calculation necessarily includes a sum over all contributions of the massive quark. To understand why this is critical, one can consider the double-real subprocess $q\bar{q} \to \ell\bar{\ell} + Q\bar{Q}$. When the heavy-quark pair is emitted in a soft and double-collinear configuration, a triple-logarithmic contribution of the form $\alpha_s^2 \ln[m]^3$ is generated. This triple logarithm is cancelled (at the level of the differential cross-section) by the exchange of a virtual soft and double-collinear massive quark-pair which is present in the two-loop form factor for a massless quark pair. Which is to say, without including all contributions involving the massive quark, the cross-section prediction will contain logarithmic sensitivity to the heavy-quark mass which is not described by universal structure of the OMEs and $\alpha_s$ decoupling relations. This is effectively a statement of the KLN theorem [117, 118].

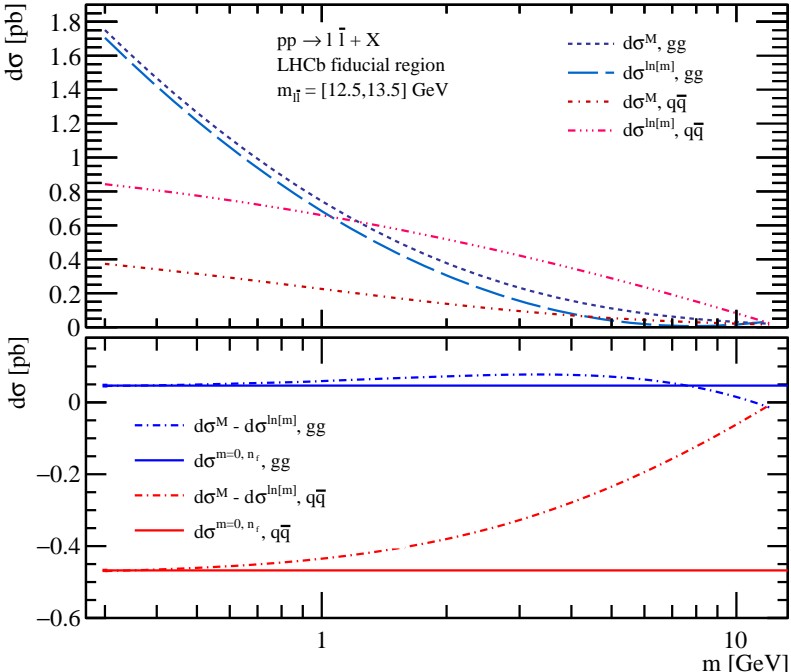

Figure 1: Upper panel: absolute cross-section for the full massive and logarithmic calculations at $\mathcal{O}(\alpha_s^2)$. Lower panel: the cross-section difference (massive-logarithmic) as compared to the direct computation of the massless $n_f$-dependent cross-section. A breakdown into partonic channels is provided.



**Extension to $\mathcal{O}(\alpha_s^3)$.** As discussed, the perturbative ingredients required to extend the procedure to $\mathcal{O}(\alpha_s^3)$ are available only for the $gg$-channel. In this case, no direct calculation of $d\sigma^{m=0,n_f}$ is available, but it can be extracted from a numerical fit to the difference $\left(d\sigma^M - d\sigma^{\ln[m]}\right)$ in the $m \to 0$ limit.

This is done by generating data for the quantity $\left(d\sigma^M - d\sigma^{\ln[m]}\right)$ for several values of $m$ in the range of $m \in [0.5, 12]$ GeV, and subsequently performing a numerical fit. By inspecting Eq. (1), the resultant distribution should be equal to the sum of the contributions from power corrections and the zero mass computation. An ansatz of the following form is therefore used for the numerical fit

$$f(m) = a_{0,0} + \sum_{i=1,j=0} a_{i,j} \left(m^2\right)^i \ln^j[m]. \tag{12}$$

The form of this ansatz is motivated by the behaviour of the squared matrix-element and phase space which both contain corrections of the form $m^2/Q^2$. The integer $j$ is limited to 2(3) when the $\alpha_s^{2(3)}$ coefficient is fitted, and a maximum value of $i = 2$ is considered in each case. The choice for $j$ is guided by the powers of collinear logarithms which may be present at each order, whereas increasing $i$ beyond 2 had little impact on the fit. The $m$-independent constant $a_{0,0}$ is equivalent to $d\sigma^{m=0,n_f}$, while the remaining terms describe the power corrections. Fitted in this way, all $m$-independent information (such as dependence on $\mu$, which is chosen as the dynamic scale $E_{T,\ell\bar\ell}$) is absorbed into the $a_{i,j}$ coefficients.

The results are shown in Fig. 2, where a total of four fitted curves are displayed corresponding to the two kinematic regimes of $p_{T,\ell\bar\ell}^{low}$ and $p_{T,\ell\bar\ell}^{high}$ at $\mathcal{O}(\alpha_s^2)$ and $\mathcal{O}(\alpha_s^3)$. In addition to the fitted central value, the fitted value of $a_{0,0}$ and its corresponding uncertainty (indicated

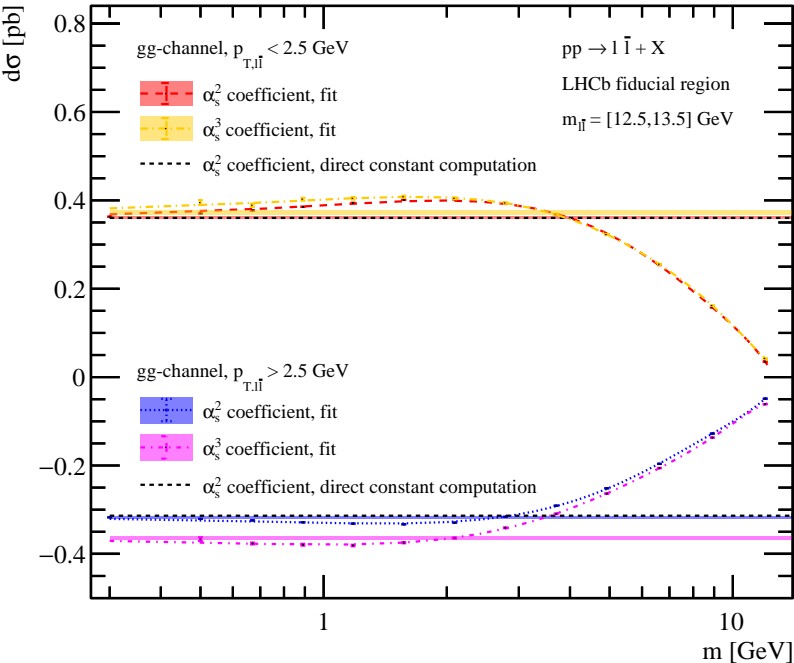

Figure 2: The same as the lower panel of Fig. 1, focussing on the $gg$-channel in two $p_{T,\ell\bar\ell}$ regions. The result of a numerical fit to the difference between the massive and logarithmic cross-sections is shown, and compared to the direct calculation of the zero mass constant at $\mathcal{O}(\alpha_s^2)$.

by a solid filled band) is displayed for each of the curves. As noted, $a_{0,0}$ should correspond to $d\sigma^{m=0,n_f}$, and it is therefore compared to the direct computation of this quantity available at $\mathcal{O}(\alpha_s^2)$. The fit leads to a result which is consistent with the direct calculation, providing confidence that the fitting procedure leads to reliable results.

From the fitted values of $a_{0,0}$ (which were produced for $b$ quarks), it is possible to construct the full $n_f$-dependent massless cross-section in the $gg$-channel. This is done by multiplying these results by a factor of $F = n_d + n_u(Q_u/Q_d)^2$, where $n_u$ and $n_d$ are the number of down- and up-type quarks. This relation holds (at the per-mille level) for the $gg$-channel as there is a direct coupling of the heavy-quark line to the gauge-boson (which is dominated by photon exchange for $m_{\ell\bar{\ell}} \in [12.5, 13.5]$ GeV). The fitted (and, where available, direct computation) are summarised in Table 1.

Table 1: Predicted and fitted values of the coefficient of the zero-mass computation up to $\mathcal{O}(\alpha_s^3)$ in the $gg$-channel. The results are for the central scale, and include an uncertainty due to the fitting procedure and the statistical error.

|  | Order | Fiducial [pb] | $p_{T,\ell\bar{\ell}}^{low}$[pb] | $p_{T,\ell\bar{\ell}}^{high}$[pb] |
|---|---|---|---|---|
| $d\sigma_{gg}^{m=0}$ | $\alpha_s^2$ | +0.51(2) | +3.96(2) | -3.45(0) |
| $d\sigma_{gg,\text{fit}}^{m=0}$ | $\alpha_s^2$ | +0.49(4) | +3.97(4) | -3.49(4) |
| $d\sigma_{gg,\text{fit}}^{m=0}$ | $\alpha_s^3$ | +0.10(6) | +4.11(5) | -4.01(4) |

**Impact of massive power-corrections.** To provide another validation of the procedure, it is also useful to directly show the contribution from the massive power-corrections $d\sigma^{pc}$ (including scale variation).

A selection of such results are shown in Fig. 3, indicating that the power corrections vanish in the limit $m \to 0$ for arbitrary scale choices. These results include those from $q\bar{q}$ and $gg$ channels (where the heavy-flavour quarks are produced in the final state) at $\mathcal{O}(\alpha_s^2)$, as well as $\mathcal{O}(\alpha_s)$ contributions from massive initial-state charm quark contributions in the $cg$ channel. In the $gg$ and $cg$ channels, the results are displayed for the $p_{T,\ell\bar{\ell}}^{low}$ and $p_{T,\ell\bar{\ell}}^{high}$ regions to indicate large cancellations which occur for the power corrections when integrated in $p_{T,\ell\bar{\ell}}$. It is worth noting that the power corrections in the $gg$-channel have a different sign at the on-shell value of the $c$- and $b$-quark mass, which leads to an additional source of cancellation. The results from massive $c\bar{c}$-initiated states were negligibly small (due to the small value of the static charm PDF), and are not shown here.

So far, the results presented as a function of $m$ are used to validate the general procedure introduced in this paper. The actual impact of the power-corrections should be studied at the values of $m = m_{c,b}^{pdf}$. As an additional note, the construction of the logarithmic cross-section should be performed in the same way that it is done for the massless calculation. For example, the logarithmic contribution generated by the PDFs and $\alpha_s$ is only present when evaluated above heavy-flavour threshold (typically $m^{pdf}$). This approach ensures that in the limit of $m = m^{pdf} \to \infty$ (where the massive cross-section vanishes), the massive power corrections obtained via Eq. (1) exactly cancel those contributions from the zero mass computation.

To place the results of Fig. 3 in context, the magnitude of the power corrections (evaluated at $m_{c,b}^{pdf}$) should be compared to that of the (total) massless computation. The results of this comparison are summarised in Table 2. The first column indicates results within the Fiducial

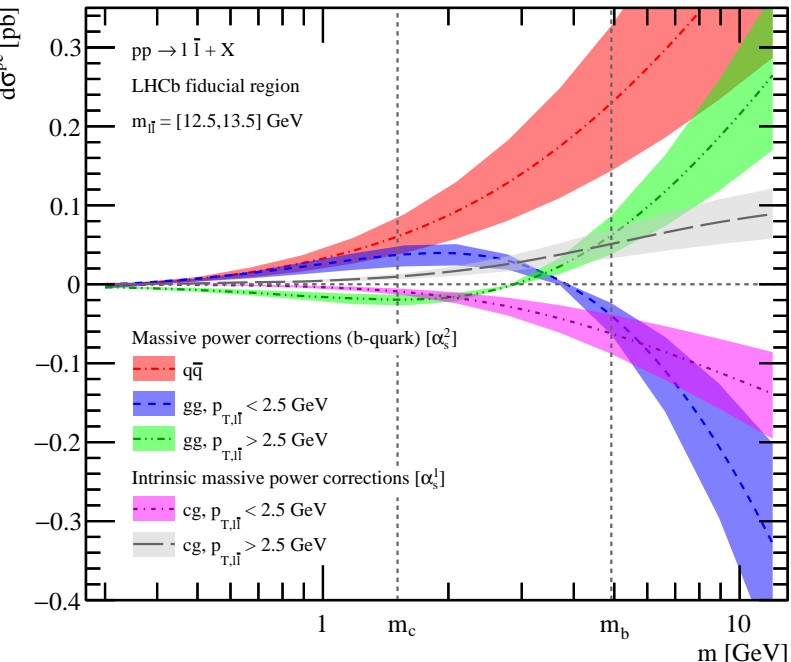

Figure 3: Massive power-corrections to the DY cross-section within the LHCb fiducial region with $m_{\ell\bar{\ell}} \in [12.5, 13.5]$ GeV. The partonic-channels, perturbative orders, and considered $p_{T,\ell\bar{\ell}}$ regions (unless inclusive) are highlighted. The scale uncertainty of each of these predictions is shown.

region, while the second column shows results in the $p_{T,\ell\bar{\ell}}^{\text{high}}$ region which includes the constraint $p_{T,\ell\bar{\ell}} \geq 2.5$ GeV. The reference calculation $d\sigma^{m=0}$ (which includes scale uncertainties) is NNLO QCD accurate within the Fiducial region, and NLO QCD accurate at finite $p_{T,\ell\bar{\ell}}$. The $\alpha_s^2$ and $\alpha_s^3$ coefficients of the massive power-corrections are shown in the same kinematic regions, where a breakdown into those contributions from charm and beauty quarks is given. Note that the charm-quark contributions in the second row are $\mathcal{O}(\alpha_s^2)$, as they include the intrinsic contributions from both the born and $\alpha_s$ coefficients. The $\alpha_s^3$ results are obtained from the functional fits shown in Fig. 2, evaluated at the on-shell mass values for each quark (with an appropriate normalisation correction for the up-type quark).

When compared to the massless prediction, the massive power-corrections introduce a correction which is typically at the level of $\approx 0.5\%$. These corrections are small in general, and there are a number of cancellations which occur between different partonic channels, different quark flavours (charm vs. beauty), and also across the $p_{T,\ell\bar{\ell}}$ spectrum (see Fig. 2 and 3). Overall, the sum of these corrections is negligible compared to the size the perturbative uncertainty of the massless calculation. Similar behaviour was found to persist for the entire $m_{\ell\bar{\ell}}$ range up to 200 GeV.

## 8   Differential distributions

This work focusses on improving our understanding the role of massive quarks in hadron-hadron scattering processes. However, in view of the on-going measurement of low-mass DY at LHCb, I also take this opportunity to provide some phenomenological results and recommendations.

As highlighted in Table 2, the overall normalisation of the massless cross-section has a

Table 2: Predictions for the DY cross-section within the LHCb fiducial region with $m_{\ell\bar{\ell}} \in [12.5, 13.5]$ GeV. The contributions from the massless calculation and the massive power-corrections are shown for the central scale. The scale uncertainties of the massless $\mathcal{O}(\alpha_s^2)$ prediction are indicated, and the uncertainties in parenthesis correspond to the fit uncertainties of the $\alpha_s^3$ coefficients.

| Prediction | Order | Fiducial [pb] | $p_{\mathrm{T},\ell\bar{\ell}}^{\mathrm{high}}$[pb] |
|---|---|---|---|
| $\mathrm{d}\sigma^{\mathrm{m}=0}$ | $\mathcal{O}(\alpha_s^2)$ | $59.9^{+2.0}_{-5.6}$ | $46.2^{+5.1}_{-8.1}$ |
| $\mathrm{d}\sigma_c^{\mathrm{pc}}$ | $\mathcal{O}(\alpha_s^2)$ | $+0.07$ | $-0.04$ |
| $\mathrm{d}\sigma_b^{\mathrm{pc}}$ | $\alpha_s^2$ | $+0.23$ | $+0.19$ |
| $\mathrm{d}\sigma_c^{\mathrm{pc}}(gg)$ | $\alpha_s^3$ | $+0.09(2)$ | $-0.04(2)$ |
| $\mathrm{d}\sigma_b^{\mathrm{pc}}(gg)$ | $\alpha_s^3$ | $+0.05(1)$ | $+0.10(0)$ |

uncertainty due to scale variation as large as 9%. It is therefore useful to consider normalised differential measurements where this theoretical uncertainty is reduced, but sensitivity to the input PDFs is retained. Such an example is

$$\frac{1}{\sigma}\frac{\mathrm{d}\sigma}{\mathrm{d}y_{\ell\bar{\ell}}}, \tag{13}$$

where $\sigma$ is the rapidity integrated cross-section for a given $m_{\ell\bar{\ell}}$ region. This observable is also experimentally well motivated as systematic uncertainties due to lepton reconstruction

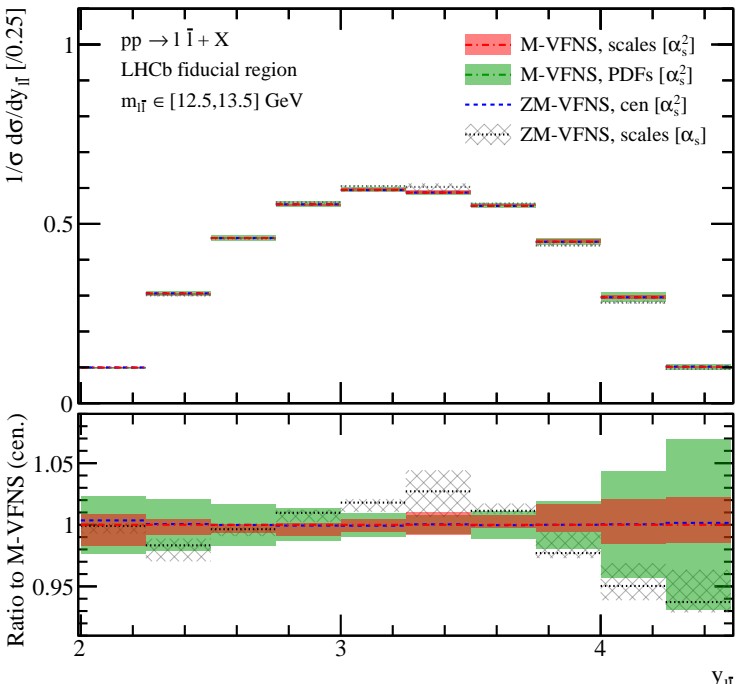

Figure 4: Differential prediction for the normalised $y_{\ell\bar{\ell}}$ distribution within the LHCb fiducial region for $m_{\ell\bar{\ell}} \in [12.5, 13.5]$ GeV.

are strongly correlated in rapidity (at fixed $m_{\ell\bar{\ell}}$). Theoretical predictions for this quantity are shown in upper panel of Fig. 4 at $\mathcal{O}(\alpha_s^2)$. In the lower panel, the various predictions and uncertainties are shown normalised to that of the central M-VFNS prediction constructed using Eq. (9). The power corrections are small, and further cancel when constructing the normalised cross-section (as the corrections are approximately flat in rapidity) resulting in a negligible contribution. The PDF uncertainties are dominant in the region of forward-rapidity, which is driven by PDF sampling in the region of small-$x$. An improved description of PDFs in this region has important consequences for neutrino astronomy [119–125], and may also provide a cross-check of those results which have been obtained using forward $D$- and $B$-hadron production data [119, 124, 126, 127]. It is therefore recommended that the experiment publishes a correlation matrix for the rapidity distributions which also includes the rapidity-integrated distribution as an entry (for a given $m_{\ell\bar{\ell}}$ region). As a final observation, the NNLO correction for this quantity is $\approx 5\%$ at large $y_{\ell\bar{\ell}}$ which may indicate the contribution of large $\ln[x]$ corrections. It could be interesting to investigate the impact of resumming these corrections, such as in [128, 129] for the DIS process, using the formalism presented in [130–133].

The theoretical study of the $p_{\mathrm{T},\ell\bar{\ell}}$ distribution is a little more delicate (particularly at large $m_{\ell\bar{\ell}}$) as a reliable description of the kinematic region of small $p_{\mathrm{T},\ell\bar{\ell}}$ relies on the resummation of Sudakov logarithms of the form $\frac{1}{p_{\mathrm{T},\ell\bar{\ell}}^2} \ln^n[p_{\mathrm{T},\ell\bar{\ell}}/m_{\ell\bar{\ell}}]$. The situation is tricky because the massive power-corrections obtained from applying Eq. (1) contain contributions which have the same form as this, and diverge in the limit $p_{\mathrm{T},\ell\bar{\ell}} \to 0$. This feature prohibits a straightforward matching of the fixed-order M-VFNS prediction with a (Sudakov) resummed calculation (and potentially also joint small-$x$ resummation [134]). This issue has been addressed and will be detailed in future work, where a dedicated study of the impact of heavy-quark mass effects on the transverse-momentum distributions of gauge-bosons will be presented.

In the region of small $m_{\ell\bar{\ell}}$ the fixed-order results are likely sufficient, and certainly useful to indicate the phenomenological (ir)relevance of the massive power-corrections. These results

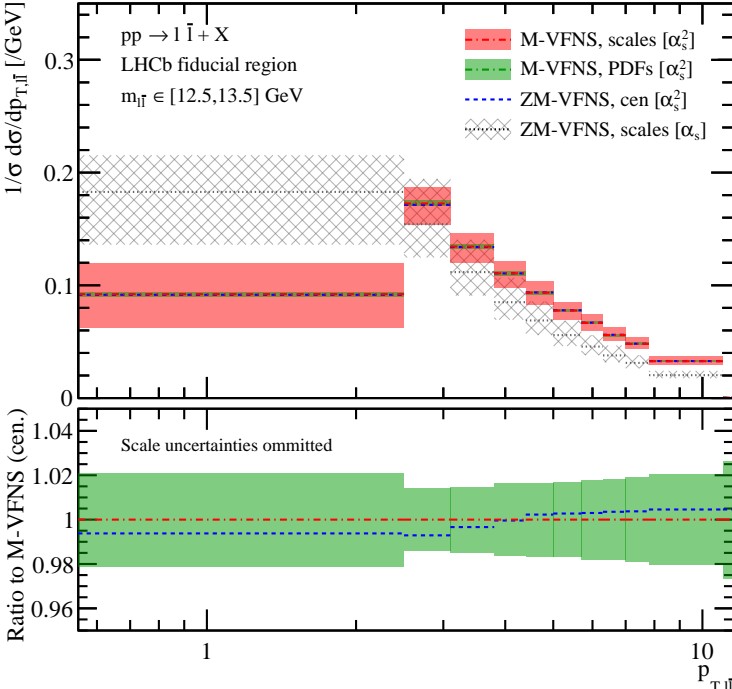

Figure 5: Differential prediction for the normalised $p_{\mathrm{T},\ell\bar{\ell}}$ distribution within the LHCb fiducial region for $m_{\ell\bar{\ell}} \in [12.5, 13.5]$ GeV.

are shown in Fig. 5 for $m_{\ell\bar{\ell}} \in [12.5, 13.5]$ GeV, where the $p_{\mathrm{T},\ell\bar{\ell}}$ distribution is shown normalised to the integrated cross-section. The impact of the massive power-corrections can be inferred by comparing the central prediction of the M-VFNS (dash-dotted red) compared to that of the massless calculation (dashed blue). The corrections amount to $\approx 1\%$ at small $p_{\mathrm{T},\ell\bar{\ell}}$, leading to a slight change in the slope of the normalised $p_{\mathrm{T},\ell\bar{\ell}}$ distribution. Overall, these effects are small compared to the either PDF or scale uncertainties (which were, for visual clarity, not shown in the lower panel).

# 9 Conclusions

The main goal of this work is to provide a deeper theoretical understanding of the treatment and role of massive quarks in predicting hadron-hadron scattering processes.

This has been achieved by studying the general structure of calculations which involve a single massive quark, and presenting a formalism to construct differential cross-section predictions in a massive variable flavour number scheme. The formalism can be applied to colour-singlet production processes as well as those involving (flavoured) hadronic jets, provided the differential observables are inclusive with respect to QCD radiation and/or are infrared and collinear safe. Hopefully, these developments will help to clarify several issues regarding heavy-quark mass effects in hadron-hadron scattering processes.

As a practical application, results have been provided for the low-mass DY rapidity distribution within the LHCb fiducial region, and I have demonstrated how a normalised distribution can give important information on the structure of the proton at low-x.

Finally, the formalism presented here represents an important step towards constructing a scheme which can be applied to predict the transverse-momentum distribution of gauge bosons, that includes the impact of heavy-flavour massive power-corrections and a resummation of Sudakov logarithhoms in a consistent way. This is a critical development towards reducing the theory systematic related to the modelling of the $p_{\mathrm{T},\ell}$ distribution in the charged-current DY process, which will in turn improve the sensitivity of LHC measurements to extract the $W$ boson mass [135].

# Acknowledgements

I am grateful to Eric Laenen for comments at the initial stages of this project. The work presented here has benefited from previous/on-going collaboration with Adrian Rodrigues Garcia, Aude Gehrmann De–Ridder, Thomas Gehrmann, Nigel Glover, and Alexander Huss as well as Marco Bonvini, Tommaso Giani and Simone Marzani. I additionally thank Valerio Bertone, Davide Napoletano, and Ben Pecjak for previous discussions about heavy-quark mass effects, and Stephen Farry and Phil Ilten for comments/correspondence about the LHCb measurement. A big thanks to Alex for providing constructive comments on this manuscript, and to the careful eyes of Lucian Harland-Lang.

**Funding information** This research is supported by the Dutch Organisation for Scientific Research (NWO) through the VENI grant 680-47-461.

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
