# Peer review of "A massive variable flavour number scheme for the Drell-Yan process"

_SciPost Physics, doi:SciPost Phys. 12, 024 (2022)_

## Round 1 · Referee Report · Anonymous (Referee 1) · 2021-8-18

Report

The paper contains the description of a new method to combine fixed-order (FO) computations performed considering the heavy-flavour quarks as massless or massive. In particular, the contributions from power suppressed terms of the form (m_q/Q)^k (where m_q is the mass of the heavy flavour and Q the typical hard scale of the process) are consistently combined with a massless computation, thereby obtaining differential distributions in a "massive variable flavour number" scheme. Such scheme allows to obtain predictions where collinear logs (log(m_q)) are resummed to all orders and exact (m_q)^k dependence is kept to the FO accuracy.

Apart from the proposed "massive variable flavour number" scheme, the paper also discusses how, using the formalism, one can numerically extract unknown results for the gg-induced DY differential cross section at order as^3.

Although I cannot claim to be a real expert on the specific topic addressed by the author and the associated literature (i.e. I have never specifically worked on the details and subtleties related to the treatment of heavy flavours in the initial state and the combination of massless with massive flavour schemes), I believe the paper contains original results.

Such results are timely too, in the sense that, as also remarked by the author, the issues discussed in the paper are likely to become more and more relevant in the years to come, in the context of precision phenomenology at the LHC.

As far as the presentation is concerned, I've found the paper well-written and clear. Similarities with previous works are properly commented, and technical points are discussed when needed, giving details at a level that I find appropriate.

For the above reasons, I recommend the paper for publication in SciPost Physics, after the following comments are addressed by the author.

  1. "Massive computation, dsigma^M" (sec. 2, page 3): isn't there also a mass-dependent contribution at order as^2 coming from "real-virtual" terms for the process pp->Z+1jet (gq ->Zq), where mass terms might arise from closed massive fermionic loops ? These terms are finite (no UV or IR divergent) and cancel for the vectorial coupling of the Z boson to the massive fermionic line, but for the axial-vector coupling the cancellation is exact only for doublets degenerate in mass (e.g. (u,d) for m_u=m_d=0), whereas a leftover scaling as (m_i)^2 - (m_j)^2 is expected to remain, m_i and m_j being the masses of the quarks belonging to the same SU(2)_L doublet (this is discussed in several places; see, for instance,

Radiative Corrections to the Ratio of Z and W Boson Production Dicus, Willenbrock Phys.Rev.D 34 (1986) 148

).

Can the author add a comment on how these terms are included (I suppose that, for consistency, they are included) ?

  1. "Logarithmic computation" (sec. 2, end of page 4): "...required to reconstruct the zero-mass limit of the massive computation". Can the author be more clear about this point? I understand it as follows:

  2. there are terms without logs, without mass dependence, and not arising from nf-type contributions (i.e. terms that don't belong to a_{0,0} in eq. 7)

  3. these terms are there also in the full massive computations

  4. eq. 3 allows one to build them and, by consequence, in the paper, such terms live in dsigma^{ln(m)}.

Is this correct, or not? Perhaps adding a further sentence to clarify this point could be useful.

  1. page 7: the sentence where the author describes how PDF uncertainty have been computed is not very clear, i.e. I don't understand, operationally, what has been done. Does the author mean that, for a given replica, dsigma_replica = dsigma_0 * K, where dsigma_0 is the prediction at order as^2 for the central set, and K would be instead disgmatilde_replica/dsigmatilde_0, where both the disgmatilde's have been computed at order as?

  2. eq. 7. I guess there's a typo: m^(2i) -> (m^2)^i ?

On a related note, can the author comment on why one starts with (m^2)^i instead of m^i ?

Always on the same equation: up to which value i and j are allowed to run in the fit ?

I understand that, physically, what matters is the log(m) dependence. However, I assume that, also in practice, a given dimensional scale mu must have been used to fit the coefficients, i.e. that, in practice, the author used log(m/mu). Is this the case ? If so, which value of mu has been used ?

Requested changes

See report for points to be addressed.

  • validity: -
  • significance: -
  • originality: -
  • clarity: -
  • formatting: -
  • grammar: -

Author:  Rhorry Gauld  on 2021-08-26  [id 1709]

(in reply to Report 1 on 2021-08-18)
Category:
answer to question

1) How are the finite corrections from virtual heavy-quark contributions proportional to the axial-vector treated?

These contributions have been included in my calculation (Refs. [92-94] of the original manuscript). This includes both double-virtual and real-virtual contributions, for all subprocesses. Practically, I define the massive power- corrections per each-quark flavour as the difference between the contribution when the calculated with either a massive or massless quark (when fixing the mass of the other quark in the doublet). Numerically these contributions were extremely small for the results shown in this paper (which are far below mll ̄ << mZ, and are therefore strongly power suppressed). For the referees interest, in the region of mll ̄ ∼ mZ these contributions are at the per-mille level for the b-quark with mb = 4.75 GeV.

I suggest to extend the citations to include reference to the work of Dicus and Willenbrock noted by theReferee. Practically, I re-calculated the real-virtual contributions myself and used the double-virtual results from the literature I had previously cited. I have now explicitly noted the inclusion of these (axial-vector) contributions in the manuscript.

2) Clarification on the ‘constant’ terms which are built by eq. (3).

The understanding of the Referee is correct, and this is certainly a tricky point. I have defined a0,0 in such a way that it is constructed from inputs that are defined in the renormalisation scheme (for PDFs and αs) where the heavy-flavour quark is massive (then built with the massless partonic cross-section). As such, these extra constant terms (i.e. the m-independent terms at the cross-section level) are absent from a0,0. However, they are part of the massive cross-section and must be constructed somehow (i.e. through knowledge of these decoupling relations for the PDFs, αs [and m its self at higher orders] computed by others in the past).

3) Definition of the PDF uncertainties.

This (the one quoted by the referee) is the definition that has been used throughout. Including a formula in the text would be appropriate to clarify this

4) The numerical fit and power corrections.

In the fit, j may run up to n where n is the order of the coefficient (i.e. 2 for αs2 and 3 for αs3). This is guided by the fact that there can be maximally these powers of collinear logarithms present (which I believe should dictates the possible powers of j present). The integer i was allowed to run up to 2, as it was found increasing this value made little difference on the fit (i.e. these higher-power terms were largely suppressed).

The ln[μ] dependence (which is certainly there, and is required from dimensional grounds as noted by the Referee) is absorbed into the constant coefficients of the fit. e.g. b_{0,1} ln[m/μ] → a_{0,1} ln[m] + a_{0,0}. I have used a dynamical scale choice μ0 = ET,ll ̄ in the numerical computation, and so practically I see this as the most obvious way to perform the fit.

Finally, concerning the choice of linear vs quadratic power-corrections in the fit. I have used (m2)i (and have corrected the typo, thank you) for the form of the power corrections, which is based on the behaviour of the squared matrix-element and phase space corrections which both contain corrections of the m2/Q2.

---

## Round 1 · Referee Report · Anonymous (Referee 2) · 2021-9-16

Report

See attachment.

Attachment

---

## Round 2 · Referee Report · Anonymous (Referee 1) · 2021-10-17

Report

I thank the author for the answers he already gave on 2021-08-26
(through the SciPost "answer to question" form), and for having
taken into account my observations in the revised version of the
manuscript.

In my opinion, now the manuscript can be published in SciPost.

---

## Round 2 · Author Response

Dear Editor, and Referees,

I would like to thank the Referees for the careful reading of the manuscript, and for providing constructive comments in various places and for making suggestions to improve the manuscript. In the following I provide a list of minor changes I have introduced to clarify various points throughout the manuscript.

With kind wishes,

Rhorry Gauld

---

## Round 2 · List of Changes

CHANGES INTRODUCED TO ADDRESS REPORT 1
Overall, four points were raised.

1a. How are the finite corrections from virtual heavy-quark contributions proportional to the axial-vector treated?
I have extended the citations to include reference to the work of Dicus and Willenbrock noted by the Referee. Practically, I re-calculated the real-virtual contributions myself and used the double-virtual results from the literature I had previously cited. I have now explicitly noted the inclusion of these (axial-vector) contributions in the manuscript.
``This includes those axial contributions arising due the presence of heavy-quark triangle diagrams, see for example [89--91]"

2a. Clarification on the ‘constant’ terms which are built by eq. (3).
I have added the following sentences to help clarify how these terms are constructed:
``Constructed in this way (i.e. using massless inputs) the logarithmic calculation will also contain those terms which are independent of $m$. They are generated by the constant terms contained in $\hat{A}_{ab}$ and $\Delta_{n_f}(\alphas)$---i.e. those which define the de-coupling across heavy-flavour thresholds in a variable flavour number scheme. It is necessary to account for these terms as they are part of the massive calculation (i.e. they appear on the LHS of Eq.~(2)), but are not generated when ${\rm d}\sigma^{\rm m=0,n_f}$ is computed with inputs (PDFs and $\alphas$) defined in the massive scheme (e.g. $n_f^{\rm max} = 4$ for the $b$-quark)."

3a. Definition of the PDF uncertainties.
I have altered the text to explicitly include the formula used to approximate the PDF uncertainties up to O(αs2):
``Where shown, PDF uncertainties have been obtained from individual replica predictions ($i$) calculated in the following way:
%
\begin{align}
{\rm d}\sigma_i = K \, {\rm d}\sigma_i[\mathcal{O}(\alpha_s)] \,,\qquad K = \frac{ {\rm d}\sigma_0[\mathcal{O}(\alpha_s^2)] }{ {\rm d}\sigma_0[\mathcal{O}(\alpha_s)] }\,.
\end{align}
%
That is to say that a differential K-factor is calculated for the central PDF member at $\mathcal{O}(\alpha_s^2)$, and then applied to each of the individual replica cross-sections which are computed at $\mathcal{O}(\alpha_s)$."

4a. The numerical fit and power corrections.
Several points regarding the value of the integers in the fit, the μ-dependence, and form of the ansatz of the power-corrections were raised. To address each these comments above (related to the fit), the text has been altered to:

``The form of this ansatz is motivated by the behaviour of the squared matrix-element and phase space which both contain corrections of the form $m^2/Q^2$. The integer $j$ is limited to $2(3)$ when the $\alpha_s^{2(3)}$ coefficient is fitted, and a maximum value of $i = 2$ is considered in each case. The choice for $j$ is guided by the powers of collinear logarithms which may be present at each order, whereas increasing $i$ beyond 2 had little impact on the fit. The $m$-independent constant $a_{0,0}$ is equivalent to ${\rm d}\sigma^{\rm m=0,n_f}$, while the remaining terms describe the power corrections. Fitted in this way, all $m$-independent information (such as dependence on $\mu$, which is chosen as the dynamic scale $E_{\rT,\Pll}$) is absorbed into the $a_{i,j}$ coefficients."

I have also changed the inline math to read $m^2)^i$ instead of $m^{2i} when defining the ansatz.

CHANGES INTRODUCED TO ADDRESS REPORT 3
The referee noted 5 places where improvements could be made.

1b. Comment about providing more detail on the construction of the logarithmic cross-section in eq. (3).
I have extended the discussion in Sec.~2, which now includes the explicit construction of the logarithmic cross-section up to $\mathcal{O}(\alpha_s^2)$, now appearing as Eq.~(5). The formula for $\Delta_{n_f}^{(i)}$ has also been given to second order in Eq.~(4)---it has been provided in the scheme where the heavy quark mass is defined in the on-shell scheme, consistent with the OME calculations.

2b. Comment about providing more detail on (previously eq. (4)) eq. (6), for the heavy quark mass slicing procedure.
The discussion appearing after eq. (4) has been extended. In particular, I have now introduced an explicit example on how the logarithmic cross-section is built at third-order (including the subtraction terms). I have also introduced an explicit discussion about the scheme dependence of the third-order results which addresses a comment in point 4. (see below) of the referee regarding the scheme dependence of the OMEs in ref.[24], and the reproducibility of the results.

3b. Validity of collinear factorisation with massive initial states.
I certainly agree that the concept of collinear factorisation is at stake, which is why Section 4 includes the statement “A deeper theoretical understanding of factorisation theorems for massive-initial states remains desirable today.” and reference to the previous work on violation of the standard factorisation theorem. The intention of this Section is to state that the procedure I have developed to extract differential massive power corrections is fully applicable to processes with massive initial states (and not to address the long-standing issue of factorisation itself). I have not introduced any changes in Section 4.

4b. Generality of the presentation of the formalism, and reproducibility of results.
The extended discussion introduced in Section 2 and Section 3 of the manuscript (see point 1. and 2. above, respectively) now addresses the comment raised by the referee about reproducibility of individual components of the calculation. The referees criticism of the approach taken by the NNPDF collaboration to extract an intrinsic charm quark PDF may be valid, but I do not believe such a discussion is relevant/appropriate/necessary within the context of my work. I also note that all the citations in the “Theoretical implementation” Section have been relevant for the current computation (either directly used, or used as a cross-check of various numerical/analytic results).

5b. Relevance of the phenomenology in Section 8.
I agree that Section 8 is out of context given the main results of the paper (which is theoretical/conceptual in nature and solves the issue of defining a massive variable flavour number scheme for differential collider observables). However, Section 8 demonstrates that the procedure is applicable to experimentally accessible distributions. These distributions have also been provided to the LHCb collaboration and will appear in a comparison to data in a forthcoming publication by the experiment. This Section is therefore still highly valuable, and remains unchanged in the revised version.

OTHER CHANGES
I have also introduced other small changes/corrections:
1c. I have included a correction in the legend (mis-label) of Figure 1 that had been made in the original submission
2c. An additional clarifying sentence in the “Numerical inputs” subsection has been included.
3c. I have included + signs before the positive values appearing in Table 1 and Table 2, and I have corrected the ”Order” for the second entry in Table 2 to now read $\mathcal{O}(\alpha_s^2)$. The latter change is necessary as is also contains the αs coefficient the way I have defined it.

---

## Editorial Decision

published